# Membrane immersion allows rhomboid proteases to achieve specificity by reading transmembrane segment dynamics

**Syed M Moin†, Sinisa Urban***

Department of Molecular Biology and Genetics, Howard Hughes Medical Institute, Johns Hopkins University, Baltimore, United States

**Abstract** Rhomboid proteases reside within cellular membranes, but the advantage of this unusual environment is unclear. We discovered membrane immersion allows substrates to be identified in a fundamentally-different way, based initially upon exposing 'masked' conformational dynamics of transmembrane segments rather than sequence-specific binding. EPR and CD spectroscopy revealed that the membrane restrains rhomboid gate and substrate conformation to limit proteolysis. True substrates evolved intrinsically-unstable transmembrane helices that both become unstructured when not supported by the membrane, and facilitate partitioning into the hydrophilic, active-site environment. Accordingly, manipulating substrate and gate dynamics in living cells shifted cleavage sites in a manner incompatible with extended sequence binding, but correlated with a membrane-and-helix-exit propensity scale. Moreover, cleavage of diverse non-substrates was provoked by single-residue changes that destabilize transmembrane helices. Membrane immersion thus bestows rhomboid proteases with the ability to identify substrates primarily based on reading their intrinsic transmembrane dynamics.

***For correspondence:** surban@jhmi.edu

†**Present address:** Vaccine Research Center, National Institute of Allergy and Infectious Diseases, National Institutes of Health, Bethesda, United States

**Competing interests:** The authors have declared that no competing interests exist

**Reviewing editor**: Christopher T Walsh, Harvard Medical School, United States

## Introduction

Signaling networks rely on the specificity of individual components for their targets, avoiding unwanted crosstalk and driving emergent properties of the system. Proteases are ubiquitous in all life, and are particularly well-adapted for serving as regulatory nodes in networks (*López–Otín and Bond, 2008*). These enzymes achieve their exquisite specificity by recognizing a short binding motif surrounding the scissile bond to align the substrate along the elongated active site cleft of the protease (*Schechter, 2005*). Residues flanking the scissile bond are designated P1 and P1′ on the amino and carboxy sides, respectively, and numbered moving outwards, while the accommodating protease subsites are correspondingly termed S1 and onwards. This geometric pairing confers high target specificity as the number of residues in the interaction increases, and results in a single and invariant substrate cleavage site (*Schechter, 2005*; *Ng et al., 2009*).

While soluble proteases have been attractive topics for study for over a century and are now understood at a sophisticated level (*López–Otín and Bond, 2008*), the more-recent discovery of intramembrane proteases uncovered a fundamentally separate path of protease evolution (*Brown et al., 2000*). These polytopic membrane proteins assemble a protease active site within the membrane using catalytic residues contributed by different transmembrane (TM) segments (*Erez et al., 2009*). Even more remarkable is the broad array of cellular networks that intramembrane proteases have come to regulate. Intensively studied is γ-secretase, an aspartyl intramembrane protease that releases signaling domains from the membrane, including the intracellular domains of the Notch receptor and the amyloid-β precursor protein (APP) implicated in Alzheimer's disease (*De Strooper et al., 1998*, *1999*;

**eLife digest** Proteases are enzymes that break the peptide bonds that hold proteins together, and have a central role in many physiological processes, including digestion, blood clotting and programmed cell death. An important characteristic of proteases is that they are highly selective, only cutting proteins that contain well-defined sequences of amino acids in accessible regions. Proteases that are soluble in water have been studied for over a century and are now well understood, as are proteases that need to be tethered to the membrane of a cell to work properly.

In 1997 researchers discovered a protease that was immersed in the cell membrane, and it soon became clear that these intramembrane proteases were widespread and involved in a wide range of processes in cells. Examples of intramembrane proteases include γ-secretase, which is implicated in Alzheimer's disease, and various site-2 proteases that regulate pathogenic circuits in bacteria.

There are many similarities between soluble and intramembrane proteases. However, given that intramembrane proteases evolved within the hydrophobic environment of the membrane, whereas soluble proteases evolved in an aqueous environment, there should also be significant differences between them. The best understood intramembrane proteases in terms of their biochemistry are probably the rhomboid proteases. However, most studies of their function have been performed in detergent systems rather than in real membranes.

Moin and Urban now report that the main strategy used by rhomboid proteases to identity the proteins that they selectively cut is completely different from that used by soluble proteases. Through a combination of biochemical and spectroscopic methods, they have discovered that rhomboid proteases identify the proteins they act on mainly by detecting changes in dynamic behavior: only those proteins that lose a stable helical structure when they exit the lipid phase to interact with the rhomboid protease will be cut. Soluble proteases, on the other hand, achieve specificity by looking for proteins with a particular sequence of amino acids. The novel strategy used by rhomboid proteases allows them to patrol the membrane for unstable helices and selectively cut them. This discovery provides the first explanation of why these complicated enzymes evolved to have active sites immersed within the cell membrane.

---

*Wolfe et al., 1999*). Homologous signal peptide peptidases function in immunity by liberating signaling domains of TNFα and FasL (*Fluhrer et al., 2006*; *Friedmann et al., 2006*; *Kirkin et al., 2007*). Site-2 proteases are metalloenzymes that release transcription factors from the membrane to regulate membrane biogenesis and stress responses in diverse organisms from bacterial pathogens to man (*Rawson et al., 1997*; *Makinoshima and Glickman, 2005*). Lastly, rhomboid proteases act as prime regulators of signaling in insects by activating EGF signals through proteolytic shedding (*Urban et al., 2001*, *2002*), and play prominent functions in diverse pathogen signaling and adhesion (*Urban, 2009*).

The presence of intramembrane proteases in all forms of life indicates that they possess a particularly useful property as regulatory enzymes (*Kinch et al., 2006*; *Lemberg and Freeman, 2007*). However, comparative approaches have been instructive only in highlighting convergent similarities in catalytic chemistry with soluble serine proteases (*Vinothkumar et al., 2010*); other properties of intramembrane proteases remain unexplored (*Erez et al., 2009*; *Urban, 2010*). Since these membrane-immersed proteases evolved within the hydrophobic milieu of the membrane, a fundamentally different environment compared to soluble proteases, could this novel environment confer different enzymatic properties? Particularly important is target specificity, because intramembrane cleavage is usually the signal-generating step that alone is sufficient for pathway activation (*Brown et al., 2000*).

With over a dozen crystal structures and well-developed reconstitution systems for their study, arguably the best understood biochemically are rhomboid proteases (*Urban, 2010*). Mutational analyses have identified some sequence determinants in rhomboid substrates, most notably small P1/P1′ residues (*Urban and Freeman, 2003*; *Akiyama and Maegawa, 2007*) and large, hydrophobic residues at P4 and P2′ (*Strisovsky et al., 2009*). However, despite the recent wealth of biochemical and structural information (*Bondar et al., 2009*; *Urban, 2010*), particularly on the *E. coli* rhomboid GlpG, most current studies have been confined to detergent systems (*Lemberg et al., 2005*; *Strisovsky et al., 2009*).

Achieving a true understanding of rhomboid function can only be realized by defining its properties in the natural context of the membrane. We therefore used biochemical and spectroscopic methods to determine the contribution of the membrane to proteolysis. These approaches revealed rhomboid proteases rely upon constraints imposed by the membrane on TM segment conformational dynamics to achieve high proteolytic specificity. Further interrogation of proteolysis directly in living cells suggest that rhomboid proteases expose the propensity of TM helices to exit the membrane and unwind as a substrate-discrimination mechanism, rather than relying on recognition-sequence binding like all other known specific proteases.

## Results

### The membrane imparts site-specificity and substrate selectivity

In order to identify any specific contributions of the cell membrane to proteolysis, we compared catalysis in living cells to catalysis in detergent micelles that support high levels of rhomboid activity. Mass spectrometry revealed that rhomboid proteolysis is notably site-specific, in contrast to other intramembrane proteases (*Fraering et al., 2004*; *Fluhrer et al., 2006*; *Friedmann et al., 2006*; *Sato et al., 2006*). In fact, cleavage of the *Drosophila* EGF ligand Spitz always proceeded between the first two residues (AS) of its TM segment even with eight diverse rhomboid proteases and in bacterial, insect and human cells (and different organelles) that harbor lipid composition differences (*Fast, 1966*) (*Figure 1A*, also see *Figure 1—figure supplement 1A*). Although Spitz is the most-studied rhomboid substrate, its cleavage site had never been mapped in cells.

Such dramatic site-specificity suggested that sequence binding positions Spitz in the active site, as occurs with soluble proteases. However, when we examined proteolysis in detergent micelles, we found that the cleavage site in Spitz also shifted +3 residues deeper into the TM segment (*Figure 1B*). The shift was even more dramatic with APP + Spi7, an engineered substrate that harbors seven TM residues of Spitz within the C-terminal 99 residues of human APP (*Urban and Wolfe, 2005*). In fact, some rhomboid enzymes abandoned the natural AS entirely in favor of cleavage +3 and/or +5 residues deeper (*Figure 1B*, also see *Figure 1—figure supplement 1B*). Without exception analyzing both N- and C-terminal cleavage products revealed that each substrate is cut only once in vitro without successive trimming, but the cut site is free to shift in position (*Figure 1B*, also see *Figure 1—figure supplement 1C*). Notably, small residues flanking the cleavage site (P1/P1′) were strongly preferred.

We found that the membrane itself is the basis for the discrepancy in site-specificity in cells vs in detergent micelles; reconstituting pure rhomboid and substrate in vitro from detergent into defined proteoliposomes restored cleavage to the natural site in Spitz, and even in APP + Spi7 (*Figure 1C*). Reconstitution into proteoliposomes comprised of a wide variety of lipids all restored site-specificity (*Figure 1—figure supplement 2*), revealing that the composition of the membrane affects the efficiency of proteolysis, but not its site-specificity. Therefore, the uncompromising site-specificity of rhomboid proteases is not an inherent property of the enzyme itself, but rather results from the membrane somehow directing the position of cleavage.

Reconstituting rhomboid and substrate into proteoliposomes from detergent micelles also revealed a second role for the membrane in substrate discrimination. Distal GA residues are a hallmark requirement for Spitz cleavage in cells (*Urban and Freeman, 2003*), and these residues were also important for cleavage in vitro when both rhomboid and substrate were reconstituted into proteoliposomes (*Figure 1D*). In contrast, cleavage of a GA mutant substrate was rescued to nearly wildtype levels in detergent micelles, suggesting that the membrane plays a direct role in restricting substrate specificity in addition to specifying the cleavage site.

### The membrane restrains rhomboid gate dynamics

Since proteolysis in detergent was notably plastic, we investigated the protein dynamics of both the protease and substrates, neither of which have yet been studied for any intramembrane protease. We functionally identified TM5 of GlpG as part of the lateral gate for substrate access to the active site (*Baker et al., 2007*; *Urban and Baker, 2008*), although other models have also been proposed (*Ha, 2009*). Since cleavage sites shifted only deeper into the TM segment, we examined gate dynamics. We introduced a nitroxide spin label onto TM5 at W236, a position we previously identified to be key for gating, and onto the overlying extramembraneous Cap loop at M247 as a control. The particular W236 and M247 sites were also attractive because neither contribute to GlpG's structural stability

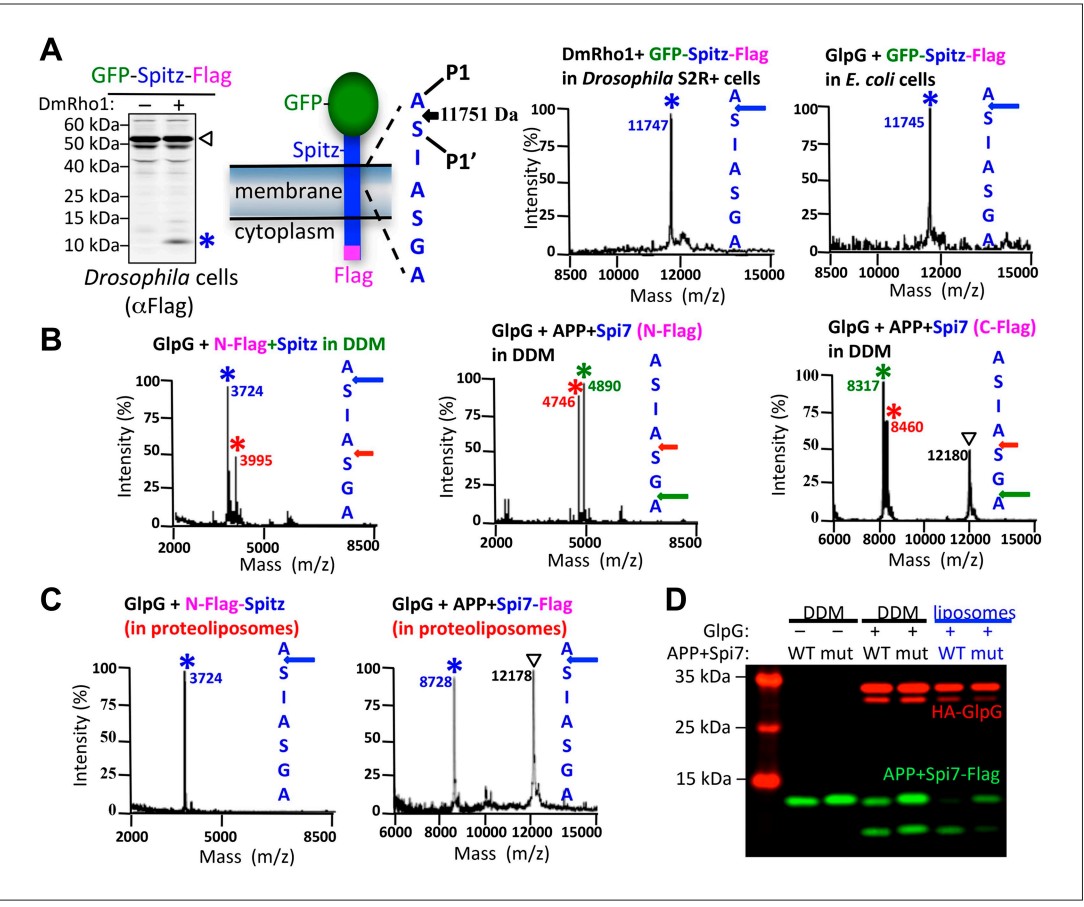

**Figure 1**. The membrane directs site and substrate specificity by rhomboid proteases. (**A**) Western analysis of GFP-Spitz-Flag expressed in *Drosophila* S2R+ cells. Denoted throughout are uncleaved (∇) and cleaved forms (*). In vivo cleavage sites were mapped by mass spectrometry following anti-flag immunocapture of GFP-Spitz-Flag processed in *Drosophila* S2R+ cells by DmRho1, as well as other rhomboid proteases in both mammalian and bacterial cells and in different organelles (also see *Figure 1—figure supplement 1A*). The invariant cleavage site is denoted with a blue arrow in Spitz (first seven TM residues are shown). (**B**) The cleavage site in Spitz generated in vitro shifted also to the second AS when assayed in dodecyl-β-D-maltoside (DDM) detergent. Arrows and asterisks are color matched throughout. Cleavage products isolated from N-Flag and C-Flag tagged APP + Spi7 substrates revealed the same cleavage sites with the expected relative proportions. (**C**) Reconstituting substrates and rhomboid proteases from detergent into proteoliposomes in vitro restored cleavage to the natural site.
(**D**) Cleavage of APP + Spi7-Flag vs its GA to LM mutant by GlpG in 0.25% DDM detergent or reconstituted into proteoliposomes. Note that upon reconstitution, the local concentration of substrate is higher than in detergent solution.

The following figure supplements are available for figure 1.

**Figure supplement 1**. Cleavage site of Spitz in animal and bacterial cells, and APP + Spi7 in vitro.

**Figure supplement 2**. Cleavage site of N-Flag-Spitz cleaved in proteoliposomes composed of different lipids in vitro.

---

(*Baker and Urban, 2012*). We then monitored dynamics directly at these sites using electron para-magnetic resonance (EPR) spectroscopy. As expected, we observed two spectral components: a dynamic form (α in *Figure 2A*) and an immobilized form (β in *Figure 2A*). These are consistent with the gate-open and gate-closed conformations, respectively, observed by X-ray crystallography (reviewed in *Urban, 2010*). Interestingly, the relative proportion of these two forms changed when GlpG was in different environments. Both TM5 and Cap sites were readily observed in the highly-dynamic form when GlpG was in the detergent-solubilized state. However, while the Cap site retained

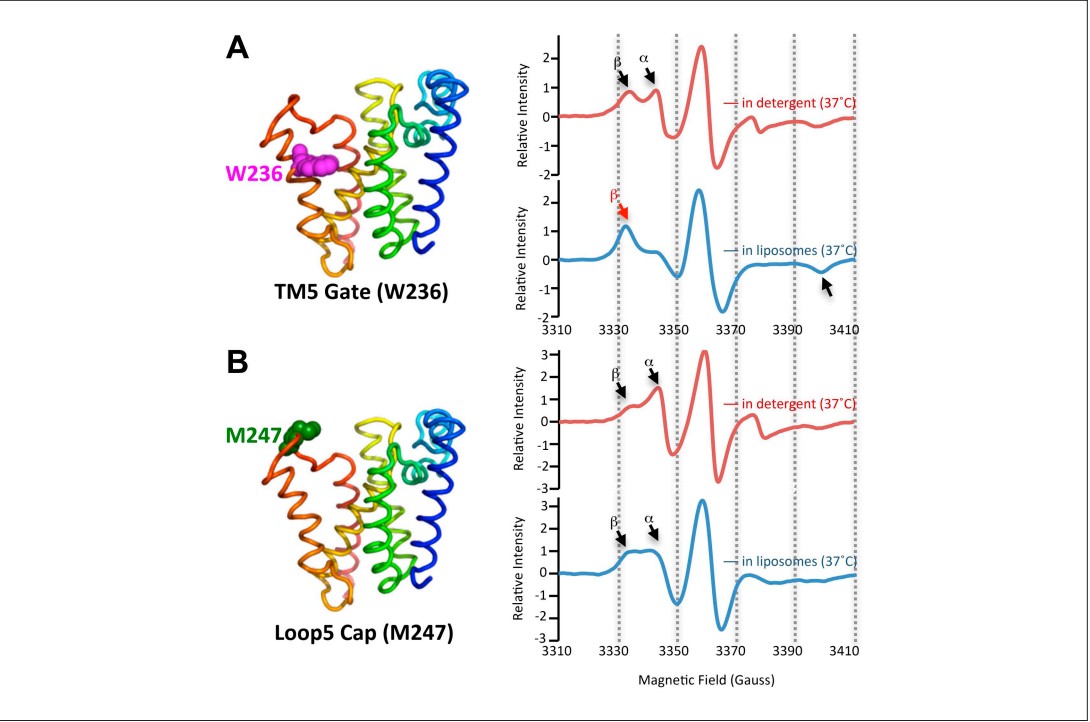

**Figure 2**. The membrane restrains rhomboid gate dynamics. Side-view of GlpG (left) showing positions (in spheres) of nitroxide spin probes. EPR spectroscopy was conducted at 37°C in 0.5% DDM detergent or proteoliposomes formed from *E. coli* lipids. Shown are 100G wide first derivative spectra with the relative signal intensity between samples normalized by quantifying the absolute number of spins. Vertical dashed lines denote magnetic field value positions. (**A**) Dynamics at the W236 gate position in DDM detergent vs proteoliposomes: note the dramatically increased amount of the immobile β component when GlpG was analyzed in proteoliposomes (red arrow) relative to in DDM detergent. (**B**) Proteins dynamics at the M247 Cap position showed a major proportion in the dynamic form when GlpG was analyzed both in DDM detergent and in proteoliposomes.

a high proportion in the mobile form when GlpG was reconstituted into proteoliposomes (*Figure 2B*), the TM5 position shifted almost completely to the strongly restrained form (*Figure 2A*). This is consistent with conversion to a predominantly gate-closed form, indicating that the membrane confers site-specificity (*Figure 1C*) by restricting gate dynamics (*Figure 2B*).

## Rhomboid substrates rely on the membrane to form stable TM helices

We next probed the structural properties of rhomboid substrates directly by examining long peptides corresponding to the entire TM segments of *Providencia* TatA, the only known bacterial substrate (*Stevenson et al., 2007*), and APP + Spi7, as well as their corresponding mutants by circular dichroism (CD) spectroscopy. CD is a powerful tool for studying TM structure, but has never been applied to the analysis of intramembrane proteolysis. Interestingly, although distal helix-destabilizing residues were required for proteolysis in proteoliposomes, both wildtype and mutant TM peptides reconstituted into proteoliposomes formed helices of indistinguishable stability as revealed by overlapping ellipticity troughs at 208 and 222 nm (*Figure 3A*). Moreover, oriented CD analysis revealed comparable spectra for both wildtype and mutant TM peptides (*Figure 3—figure supplement 1*), suggesting that the tilt of the substrate and uncleavable TM segments in the membrane is similar. The knowledge-based Ez algorithm (*Schramm et al., 2012*) also failed to detect any tilt or structural differences between these wildtype and mutant TM segments.

However, the key difference was that rhomboid substrates actually rely on the membrane to form these stable helices. TatA was 31% less helical, and APP + Spi7 over threefold less helical, in DDM micelles compared to being reconstituted into proteoliposomes (*Figure 3A*). This observation explains the relaxed need for helix-destabilizing residues for cleavage in detergent (*Figure 1D*), because these

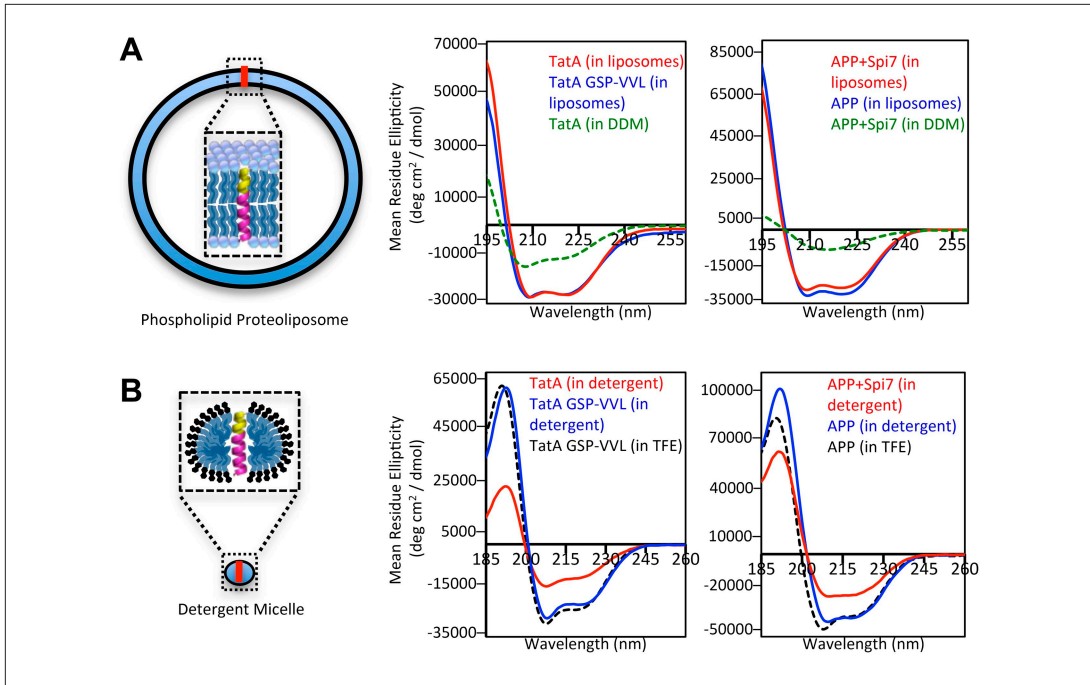

**Figure 3**. The membrane restrains substrate TM dynamics. (**A**) CD spectroscopy of substrate (red traces) and non-substrate (blue traces) TM peptides revealed both display similarly stable helices when reconstituted into proteoliposomes, which was dramatically higher than in DDM detergent micelles (green dashed traces). (**B**) CD spectroscopy of substrate TM peptides in detergent micelles (red traces) revealed them to be strongly reduced in helicity compared to non-substrates (blue traces); in comparison, non-substrates formed helices in detergent micelles similar in stability to those in the helix-inducing solvent TFE (black dashed traces). All values are mean residue ellipticity, with relative peptide concentrations determined by simultaneously monitoring actual peptide bond absorbance during CD scanning. The TatA GSP-VVL mutant is G11V + S12V + P13L.

The following figure supplements are available for figure 3.

**Figure supplement 1**. Oriented CD Spectroscopy of wildtype and mutant APP and TatA TM segments in phospholipid bilayers.

TM segments are already partly unwound. Moreover, we found that this instability is a defining feature of rhomboid substrates relative to non-substrates; analysis in detergent micelles revealed that the APP + Spi7 peptide was a remarkable 36% less helical than the non-substrate APP, while TatA was 46% less helical than its uncleavable GSP-VVL mutant (**Figure 3B**).

Spectroscopic interrogation revealed for the first time that rhomboid substrates are unable to maintain a stable helix without the membrane, raising the possibility that differences in intrinsic TM dynamics first and foremost is the property that defines substrates, rather than serving a secondary role in exposing the substrate backbone for hydrolysis as currently thought (**Ha, 2009**; **Strisovsky et al., 2009**). This dynamic nature could allow substrates, but not non-substrates, to enter the catalytic center for proteolysis to ensue. While this model explains our biophysical observations, it's based entirely on in vitro measurements. We therefore sought to test the physiological, as well as functional, relevance of our model in living cells by examining its central predictions.

## TM dynamics, not sequence binding, position substrates in the active site

The distinguishing prediction of our model is that substrate position in the active site is dictated primarily by protein dynamics caused by residues of the TM segment that, when not supported by the membrane, disrupt helical stability and promote entry into the hydrophilic active site. Substrate position should therefore be predictably shifted by moving, enhancing, or limiting intrinsic substrate dynamics. In contrast, if rhomboid achieves specificity by regimented binding of a recognition motif

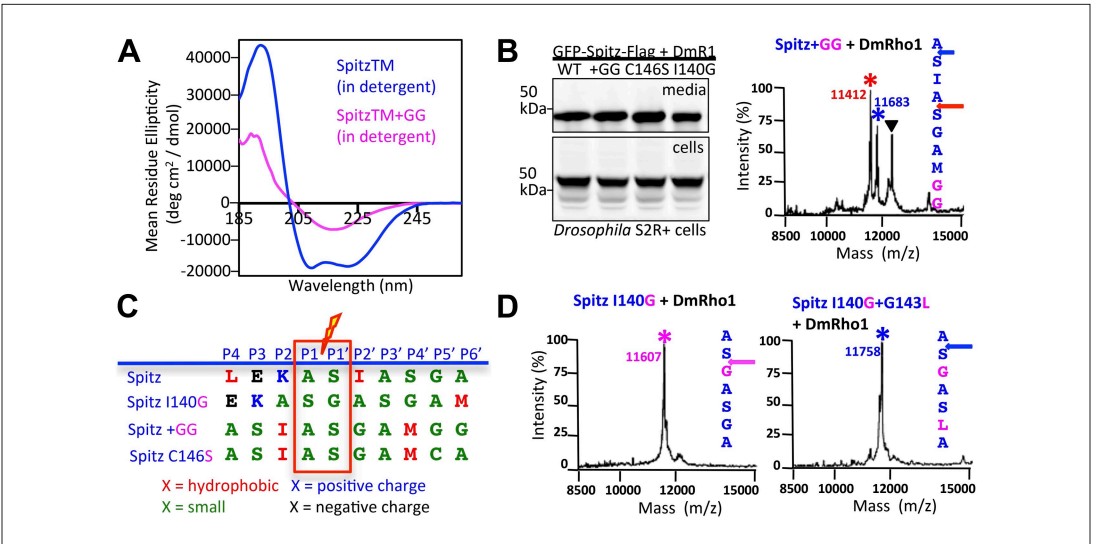

**Figure 4**. Substrate dynamics position the cleavage site by *Drosophila* rhomboid-1 in living cells. (**A**) CD spectroscopy revealed that incorporating two glycines (+GG) near the middle of the Spitz TM further reduces its helical stability. (**B**) Western analysis of Spitz mutant processing by DmRho1 in living *Drosophila* cells assessed by amount of Spitz released into the media (top left panel). In vivo cleavage sites of Spitz + GG revealed a +3 site shift in cleavage. Black triangles denote IgG light chain from the immunoisolation. (**Figure 4—figure supplement 1** shows cleavage of Spitz C146S, and a +5 site shift in cleavage of Spitz + GG by GlpG in *E. coli* cells). (**C**) Alignment of different Spitz mutant cleavage sites generated by DmRho1 in living cells. All mutants were cleaved efficiently (in **B**), yet note the dramatic redistribution of residues at P4, P3, P2, and P2′ positions as cleavages sites shifted. (**D**) The cleavage site of the Spitz I140G mutant shifted +1 residues with *Drosophila* rhomboid-1. Mutating the distal, helix-destabilizing G143 residue to leucine shifted cleavage site −1 residues outwards.

The following figure supplements are available for figure 4.

**Figure supplement 1**. TM protein dynamics position the cleavage site.

like soluble proteases, then the site of cleavage must necessarily be fixed, while altering TM dynamics should only change the efficiency of cleavage. We tested this prediction with *Drosophila* rhomboid-1 (DmRho1), the natural Spitz protease, and *E. coli* GlpG, the best understood rhomboid protease, under physiological conditions. To reveal substrate position in the active site at the time of catalysis we mapped substrate cleavage sites generated in living cells, since a protease:substrate complex structure has never been achieved.

First, we reasoned that we could increase substrate TM dynamics but leave unperturbed all residues thought to be important for Spitz 'binding' (P4–P2′) by mutating two residues near the middle of the Spitz TM segment (at P8′ and P9′) to glycine. Circular dichroism spectroscopy revealed that the two glycines dramatically decreased helical stability by >60% relative to wildtype Spitz (**Figure 4A**). Next, we tested Spitz + GG cleavage and found it to be cleaved more efficiently than wildtype Spitz in *Drosophila* cells, but the cleavage site shifted +3 residues (**Figure 4B**). The cleavage site also shifted in living *E. coli* cells by GlpG (**Figure 4—figure supplement 1A**). This effect was not limited to glycine, since changing a distal cysteine (at P8′) to a helix-destabilizing serine also resulted in very efficient cleavage, but again induced a +3 residue shift in cleavage site (**Figure 4—figure supplement 1B**). These shifts are particularly instructive because the natural sequence containing any possible binding motif is completely unperturbed, yet is abandoned for an 'impermissible' sequence (**Strisovsky et al., 2009**) with alanine and glycine at P4 and P2′, respectively (**Figure 4C**). Moreover, addition of helix-destabilizing residues at other positions could also induce shifts: mutating the P2′ isoleucine to glycine caused a +1 shift (**Figure 4D**).

Second, since we found site-specificity and gate dynamics were inversely correlated, we next examined whether gate-open mutants cause cleavage site shifts under physiological conditions. Indeed, all

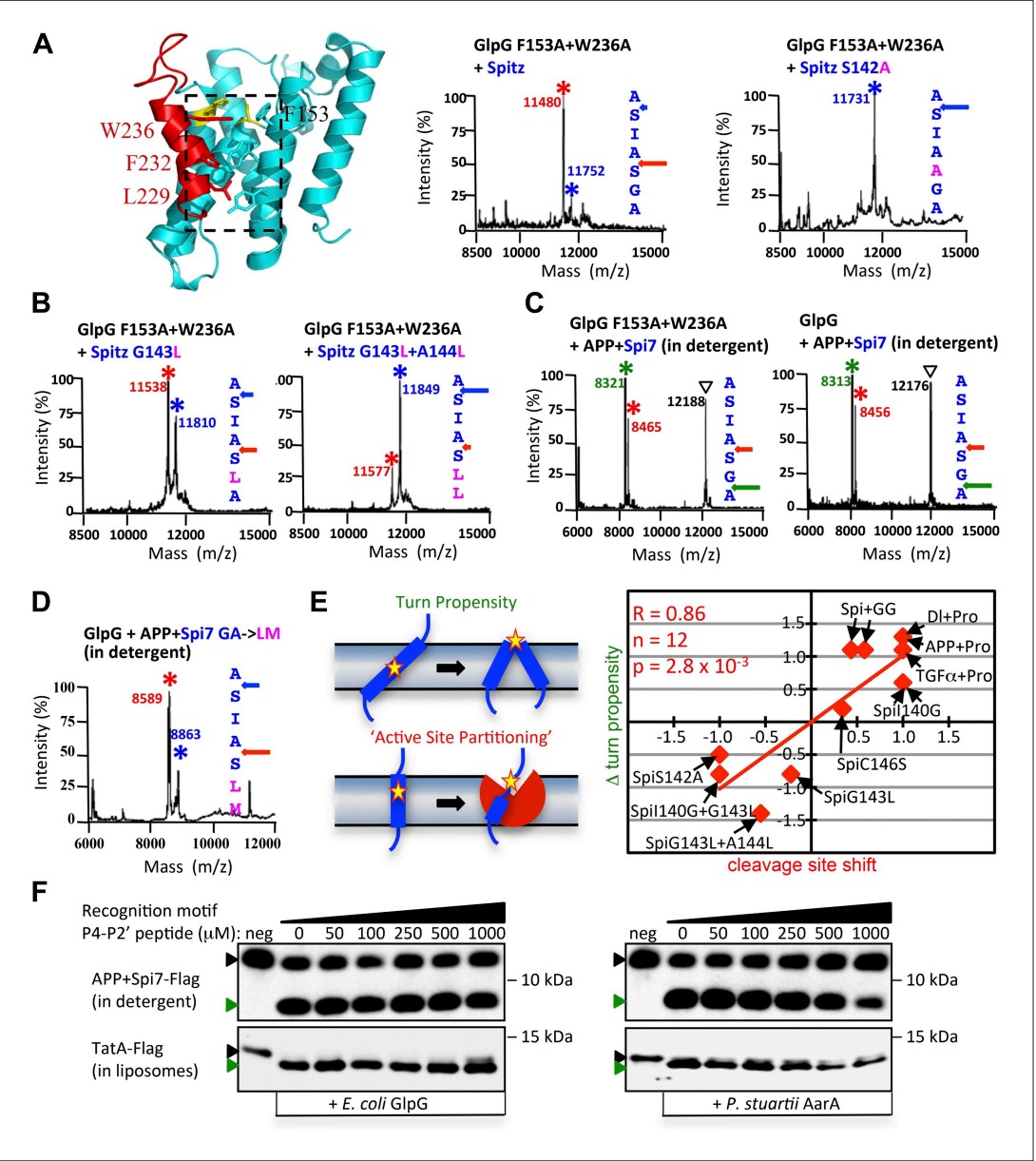

**Figure 5**. Rhomboid gate and substrate dynamics position the cleavage site by bacterial rhomboid proteases.
(**A**) Lateral view of GlpG showing TM5 interfacing sidechains (boxed) whose mutation opens the substrate gate (red) and increases protease activity. Gate-open mutants of GlpG analyzed in *E. coli* shifted cleavage site deeper into the Spitz TM segment (red). Cleavage of Spitz with helix-destabilizing S142 mutated to alanine by gate-open GlpG in *E. coli* cells produced a complete shift towards the top of the TM segment (blue). (**B**) Cleavage of Spitz with G143, and G143 + A144, mutated to leucine by gate-open GlpG in *E. coli* cells produced a gradual shift in cleavage site outwards. (**C**) Wildtype and gate-open (F153A + W236A) GlpG proteases produced identical, deeper cleavage sites when assayed in detergent micelles, indicating that the gate is fully open in the absence of the membrane. (**D**) Mutating the distal GA residues to LM also shifted the cleavage of APP + Spi7 in DDM detergent micelles to the natural AS site. Under these conditions the gate of wildtype GlpG is 'open' by virtue of the membrane being absent. (**E**) Diagram illustrating the 'turn propensity' effect of incorporating a helix-destabilizing/membrane-exiting residue (asterisk) into a long TM segment (***Monné et al., 1999b***). Lower diagram proposes an analogous effect for intramembrane proteolysis: residues of high 'turn propensity' could promote lateral substrate partitioning into the rhomboid active site. Right: change in turn propensity of substrate mutants plotted against the change in cleavage site occurring in natural membranes of living cells. (**F**) A hexapeptide encompassing the entire recognition motif (P4–P2') of *P. stuartii* TatA, the most efficient rhomboid substrate, failed to block cleavage of two different substrates by any rhomboid tested in detergent or reconstituted
*Figure 5. Continued on next page*

*Figure 5. Continued*

liposomes (see *Figure 5—figure supplement 2* for APP + Spi7 cleavage in liposomes and TatA cleavage in detergent). Black and green triangles denote substrate and cleavage product bands, respectively. The highest tested peptide concentration was 1 mM, while substrates were maintained at ≤1 µM.

The following figure supplements are available for figure 5.

**Figure supplement 1**. Cleavage site shifts with gate-open mutants.

**Figure supplement 2**. Poor competitive inhibition of proteolysis by a 1000-fold excess of a recognition motif peptide.

gate-open GlpG mutants (*Baker et al., 2007*; *Urban and Baker, 2008*) analyzed in living *E. coli* shifted the Spitz cleavage site +3 residues deeper into the TM segment (*Figure 5A*). The effect of these mutations is unlikely to be explained by interfering with any putative recognition-binding site on the protease, because all gate-open mutants, irrespective of position, resulted in cleavage site shifts (*Figure 5—figure supplement 1A*). In fact, wildtype GlpG in detergent produced the same +3 and +5 cleavage sites as gate-open mutants (*Figure 5C*, also see *Figure 5—figure supplement 1B*), suggesting that, in the absence of the membrane, even the wildtype gate opens fully (which is also independently supported by EPR analysis).

Third, we 'limited' substrate dynamics, which our model predicts should decrease substrate 'reach' into the rhomboid active site, and thereby shift cleavage to the top of the substrate TM. Strikingly, the cleavage site of Spitz by gate-open GlpG in vivo shifted completely −3 residues when helix-destabilizing S142 alone was replaced by alanine (*Figure 5A*). Moreover, a partial −3 shift also occurred when we replaced G143 with leucine, and almost fully to the outer AS when both GA residues were replaced with leucine (*Figure 5B*). A shift also occurred with *Drosophila* rhomboid-1 (*Figure 4D*). Importantly, we observed the same shifts with mutant substrates and wildtype GlpG in detergent micelles, in which the gate is open by virtue of the membrane being absent (*Figure 5D*). Therefore, even when the gate is open, helix-destabilizing residues are required for substrates to enter rhomboid's active site.

## A 'helix-and-membrane-exit' propensity scale correlates with cleavage site shifts

Although we made as conservative mutations as possible with respect to size when altering TM dynamics, mutants should be interpreted with caution because they can also have unintended effects, including altering the TM surface and/or interface with rhomboid. To evaluate further what physical property is most likely responsible for the effects on proteolysis, we searched for a correlation between cleavage site shifts in living cells vs changes that our mutants made in residue properties using several independently generated physical scales. We found the strongest correlation with a 'TM turn propensity' scale (*Monné et al., 1999a*, *1999b*). This scale had been derived by placing guest residues into the center of a long TM segment and measuring whether the residue prefers to stay in the middle of the membrane in the TM helix, or moves outside the membrane in a turn that breaks the long TM helix into two shorter TM helices (*Figure 5E*). The quantified propensity scale displayed a correlation coefficient of 0.86 with the direction and degree of cleavage site shifts that we observed with 12 mutant substrates cleaved by rhomboid proteases in cellular membranes (*Figure 5E*).

Taken together, these observations, conducted in living cells, reveal that gate and TM dynamics, rather than binding of a specific sequence within Spitz, play the predominant yet completely overlooked role in positioning substrates into the active site. The new cleavage sites further revealed dramatic re-alignments of the substrate in the rhomboid active site (*Figure 4C*), even when the natural putative recognition was unperturbed, such that stereotypical binding of a complementary recognition sequence between rhomboid and substrates is unlikely to be the main driving force for protease specificity. But to evaluate this further, we examined the ability of the proposed P4–P2' recognition binding region of TatA, the most efficient bacterial substrate, to inhibit proteolysis competitively

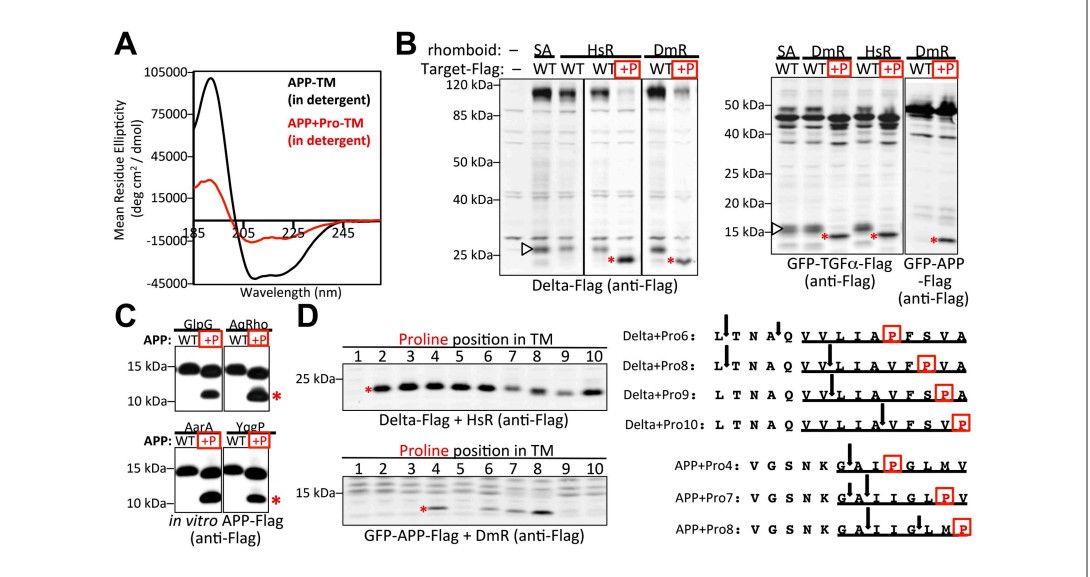

**Figure 6**. A single proline converts non-substrates into rhomboid substrates. (**A**) CD spectroscopy revealed that incorporating a single proline into the TM of APP (at position 7) dramatically reduces its helical stability. (**B**) Incorporating a single proline into the TM segment of Delta, TGFα, or APP converted each into an efficient substrate for rhomboid proteases (DmR is *Drosophila* rhomboid-4, HsR is human RHBDL2, SA is the catalytic serine mutant of DmR). (−) shows non-specific anti-Flag bands in untransfected HEK293 cells. The natural juxtamembrane cleavage product generated by metalloprotease shedding (denoted by a white triangles) was outcompeted by intramembrane cleavage. (**C**) Incorporating a single proline also converted APP-Flag into an efficient substrate in vitro with pure bacterial rhomboid enzymes (cleaved products denoted with asterisks). (**D**) Effect of proline position on rhomboid proteolysis (shown are cleaved product bands, highlighted with asterisks). Cleavage sites of Delta + Pro-Flag and GFP-APP + Pro-Flag with different proline positions were mapped from living HEK293 cells (right panel). TM segment residues are underlined and the exogenous proline is boxed.

The following figure supplements are available for figure 6.

**Figure supplement 1**. Induction of non-substrate cleavage by rhomboid proteases.

---

by a panel of bacterial rhomboid enzymes in vitro both in detergent micelles and reconstituted into proteoliposomes. Even at millimolar concentration (~1000× the substrate concentration), the peptide did not block rhomboid proteolysis of TatA or APP + Spi7 substrates (***Figure 5F*** and ***Figure 5—figure supplement 2***), independently suggesting that sequence-specific binding is not the main feature of rhomboid specificity.

## Increasing TM dynamics converts non-substrates into rhomboid substrates

Discovering rhomboid specificity is driven by exposing intrinsic TM dynamics raised an independent prediction: non-substrates of various sequence should be converted into substrates simply by increasing their intrinsic TM dynamics. APP, Delta and TGFα have been characterized genetically, cell biologically, and in vitro with pure proteins as non-substrates for rhomboid enzymes (***Peschon et al., 1998***; ***Urban and Freeman, 2003***; ***Lemberg et al., 2005***; ***Urban and Wolfe, 2005***; ***Adrain et al., 2011***). We reasoned that incorporating a single proline, which has the highest helix-and-membrane exit propensity of all residues (***Monné et al., 1999a***, ***1999b***), might increase their intrinsic TM dynamics. Indeed, CD analysis revealed that incorporating a single proline into the TM of APP was sufficient to decrease TM helix stability by over twofold (***Figure 6A***). Remarkably, installing single prolines alone was sufficient to convert APP, Delta and TGFα into efficient rhomboid substrates in vivo and in vitro (***Figure 6B,C***). In fact, intramembrane cleavage become so efficient that in all cases it outcompeted the natural juxtamembrane cleavage by metalloproteases, and in the case of Delta,

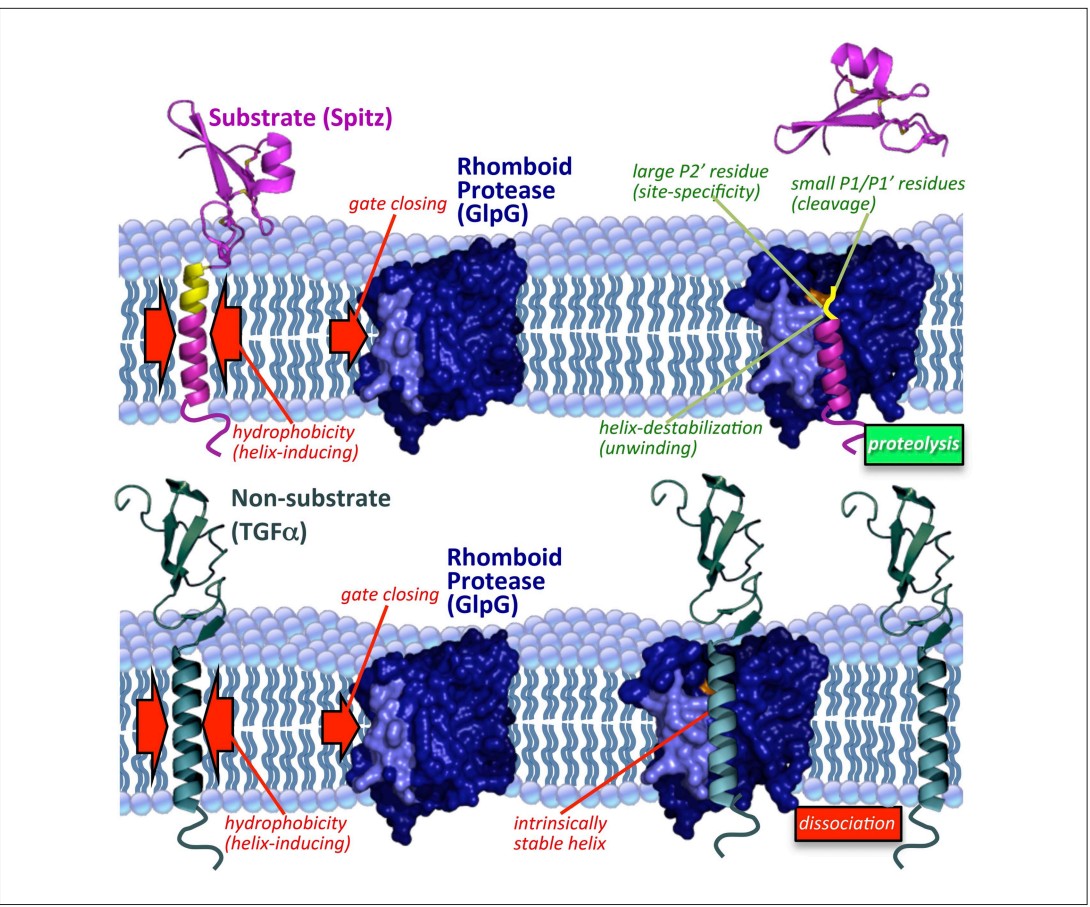

**Figure 7**. Model of rhomboid proteolysis driven by intramembrane protein dynamics. The membrane imposes two constraints on protein dynamics to ensure high proteolytic specificity; it induces helix formation of TM segments (left red arrows) and limits rhomboid gate (light blue) opening (right red arrow). Substrates form a stable helix only in the membrane; partial exposure to the aqueous environment within rhomboid triggers an entropy-driven conformational switch, facilitated by helix-destabilizing residues, allowing substrates to reach the catalytic residues (in orange). Bottom panel depicts a non-substrate:rhomboid complex, in which the TM segment maintains a stable helix and therefore cannot reach the catalytic residues. The non-substrate TM segment eventually dissociates without being cleaved (far right). Induction of efficient non-substrate cleavage suggests that the initial docking interaction between rhomboid and TMs is non-specific. The exact order of events, and what triggers each step, remain speculative. Membrane thinning surrounding GlpG as observed in molecular dynamics simulations is illustrated (**Bondar et al., 2009**; **Zhou et al., 2012**), although its functional consequence remains unclear. Structures 2IC8 (closed GlpG), 2NRF (open GlpG), 1MOX (Spitz-EGF), and 2TGF (TGFα-EGF) were used to diagram the model.

little unprocessed full-length protein could be detected. Cleavage was dependent on rhomboid activity, because it required the catalytic serine of both *Drosophila* and human rhomboid proteases, and γ-secretase inhibition did not affect cleavage (**Figure 6—figure supplement 1**).

By scanning the proline into each of the first 10 TM positions, we found multiple sites and surprising sequence diversity could accommodate rhomboid cleavage. Although incorporating proline in any TM position between 2 and 10 resulted in Delta and TGFα cleavage, APP displayed a more limited profile of acceptable proline site position. This may reflect influence by neighboring sequence context on overall helical stability and/or propensity to partition into the active site (**Li et al., 1996**; **Yang et al., 1997**). Intriguingly, the relative cleavage site preference was dependent on the position of the proline, and moved deeper as the proline position descended into the TM segment (**Figure 6D**). These observations provide compelling evidence that simple changes in intrinsic TM dynamics under physiological conditions drive substrate specificity for membrane-immersed proteolysis.

## Discussion

### Exposing intrinsic TM dynamics is a novel proteolytic specificity mechanism

Immersion of a hydrolytic reaction within the hydrophobic membrane has long been studied as a biochemical conundrum (*Erez et al., 2009*), yet the possible advantage of this arrangement has not been explored. We integrated diverse analytical approaches including spectroscopy, defined reconstitution systems, and cleavage site mapping in living cells to probe the role of the membrane. These new approaches ultimately converged to reveal that the main property conferred by membrane-immersion is the ability to identify substrates through a mechanism centered on exposing intrinsic TM dynamics instead of the protein-protein binding strategy used by other specific proteases (*Figure 7*).

Applying CD spectroscopy provided the first opportunity to measure the structural properties of rhomboid substrate TMs relative to non-substrates, and in different environments. Coupling this information with examining proteolysis in living cells suggests that the main factor keeping many type I TM segments from becoming rhomboid substrates is their ability to maintain a stable TM helix. Differences between substrate and non-substrate TM helices are minimal while they reside in the membrane. However, a defining property of rhomboid substrates is a meta-stable TM helix that actually relies on the membrane for stability, and unravels without it. Mutations that stabilize TM helices compromise proteolysis, while simply introducing helical instability into non-substrate TMs, as measured directly by CD, converted three of three unrelated type I membrane proteins into substrates for seven different rhomboid proteases despite diversity in the cleaved sequences.

It is worth emphasizing that mutational analysis has played an important role in our study of rhomboid proteolysis, and while this is a proven strategy for elucidating enzyme mechanisms, all of our mutants that change helicity also necessarily change sequence. To limit this inherent drawback, we were careful, whenever possible, to make conservative mutations, measure effects on helicity directly by CD, and ultimately base our analysis on >30 mutations. Nevertheless, we further evaluated what underlying physical property drives the cleavage site shifts and ultimately found the strongest correlation with a 'turn propensity' scale. The informative feature of this novel scale is that it integrates both the helical propensity of a residue, as well as its preference to be inside the membrane versus seeking a more hydrophilic environment (*Monné et al., 1999a*, *1999b*). As such, the intrinsic dynamics in rhomboid substrate TMs thus likely derives from a combination of residues that destabilize the TM helix structure directly, as well as those of limited hydrophobicity that increase the likelihood of this polypeptide region escaping the helix-inducing environment of the membrane (in favor of the hydrophilic active site of rhomboid, see below). This observation independently suggests that the key differences in TM dynamics are evaluated in a different, non-membranous environment. This may explain why alanines are known to be important for rhomboid proteolysis yet themselves are not helix-destabilizing directly, and require mutation to large hydrophobic residues to compromise proteolysis.

The strong correlation with the turn propensity scale could also have predictive value for finding new rhomboid substrates by sequence analysis. However, it should be noted that our analysis had the benefit of quantifying 'changes' in turn propensity ('Δ turn propensity' in *Figure 5E*) between two sequences that differ at only one or two residues. In practice, the turn propensity of a residue has been shown to be altered in non-linear ways by differences in TM segment length, the local sequence context, and the exact position of the residue within the TM segment (*Monné et al., 1999b*). If these confounding parameters also apply to rhomboid proteolysis, accurately predicting absolute propensities of natural TM segments from sequence alone may present challenges. Nevertheless, cautious optimism is warranted, since these challenges could be overcome by additional scale refinement with rhomboid proteases directly.

While our analyses consistently indicate TM dynamic state is the defining feature of rhomboid substrates, they do not neglect that sequence also contributes a secondary role. The shifts that we observed reveal sequence requirements for proteolysis with positive data, because they mark substrate position in the active site at the time that catalysis was proceeding efficiently. Preference for cleavage to shift to between small residues re-affirmed prior studies (*Urban and Freeman, 2003*; *Akiyama and Maegawa, 2007*). However, little appears to be essential for cleavage beyond small P1/P1′ residues, since we found a great diversity of residues at other positions allowed efficient proteolysis to proceed by multiple rhomboid proteases. This is particularly informative, because prior

analyses necessarily focused on mutations that block cleavage, but could not rule out the possibility that they interfere with proteolysis indirectly, for example, by promoting TM oligomerization (which is required for TatA function).

By incorporating a single proline at various positions we were able to convert three unrelated proteins into rhomboid substrates. Nevertheless it is important to note that this too does not mean that any TM sequence can become a rhomboid substrate. Rather, it highlights that the natural sequence diversity in many TM segments provides ample opportunities for finding acceptable cleavage sites. True substrates would nevertheless be expected to have further sequence optimization that would be specific to a particular rhomboid protease, because greater proteolytic efficiency would be favored as substrate and protease co-evolve. One current example might be TatA proteolysis, because TatA cleavage is thought to proceed rapidly and automatically for quorum sensing to operate (*Stevenson et al., 2007*). However, this is unlikely to be representative of most rhomboid functions. Even so, a peptide comprised of all P4–P2' residues requires millimolar concentrations to affect proteolysis of TatA by AarA, indicating that even in this context recognition sequence binding is not the main determinant for specificity.

Considered together, our observations suggest a new working model in which canonical rhomboid proteases patrol the membrane for meta-stable TM helices for cleavage. A gate-open rhomboid may be sufficient to provide a microenvironment in which 'intrinsic' TM segment differences can be unmasked, and recent molecular dynamics simulations and biophysical measurements indicate that this is a stable rhomboid conformation in the membrane (*Baker and Urban, 2012*; *Zhou et al., 2012*). Because of both helix-destabilizing residues and limited hydrophobicity, substrates are poised to exit from the membrane and partition into the hydrophilic rhomboid active site. Proteolysis then ensues because such extended TMs are susceptible to proteolysis, and/or are able to 'reach' the internal catalytic apparatus. Such substrate 'partitioning' is consistent with a strong correlation with a 'turn propensity scale', and may further explain the unexpected observation that scanning a single proline along the TM segment moves the cleavage site deeper, at times positioning it at unacceptable residues. Importantly, the central yet overlooked component of this specificity system is the membrane itself, which limits proteolysis both by inducing TMs to form helices and restricting gate-opening (*Figure 7*).

Ultimately our analyses have both coalesced into a general framework for how rhomboid intramembrane proteolysis functions, and emphasize that further work is required to define the specific details of how this complex system operates. Differences between substrates and non-substrates in the membrane beyond the resolution of our experimental approaches are possible, although direct evidence suggests that the most dramatic difference occurs when TMs leave the influence of the membrane. Moreover, structural analysis of the rhomboid-substrate complex is required to define the fine-detail interactions that mediate proteolysis, although this might prove particularly challenging if true substrates are indeed intrinsically dynamic. Finally, the precise order of events, and what triggers each step in the cleavage reaction, also remain unclear.

## Concluding perspective

It's well recognized that proteolytic release of factors from the membrane regulates many signaling networks. Yet dozens of examples reveal that simply anchoring a protease domain to the membrane through a TM segment satisfies these needs (*Blobel et al., 2009*; *Antalis et al., 2010*). So why did complicated, membrane-immersed enzymes evolve and become so wide-spread across all forms of life? Our analyses indicate that rhomboid proteases achieve substrate specificity first through reading TM dynamics, which endows them with different properties relative to soluble proteases. Intramembrane proteases might therefore be a distinct group of enzymes in the cell, not because they are proteases that release proteins from the membrane, but because they are membrane-immersed; they live in a world with different rules, giving the cell a set of enzymes with unique properties that it can harness for evolving new functions.

## Materials and methods

### In vitro proteolysis assays

Rhomboid proteases and substrates were expressed in *E. coli,* purified, and cleavage reactions were conducted at 37°C for 1–2 hr in 50 mM Tris–HCl pH 7.4, 150 mM NaCl, and either reconstituted into proteoliposomes or in 0.1% DDM as described previously (*Urban and Wolfe, 2005*). Reaction

products were resolved and quantified by infrared fluorescence (LiCor Biosciences, Lincoln, NE) using western analysis (*Baker et al., 2007*).

## Mass spectrometry

Substrates from in vitro proteolysis assays or transformed/transfected cells (lysed in RIPA buffer) were subjected to anti-Flag immunoaffinity purification with the M2 resin (Sigma, St Louis, MO), and analyzed by MALDI-TOF mass spectrometry using sinapinic acid as the matrix as described previously (*Baker et al., 2007*).

## Propensity analysis

Changes in turn propensity of substrate mutants was calculated by subtracting the average turn propensity value (from table 1 in *Monné et al., 1999b*) of the wildtype residue from the mutant residue. This value was plotted against the change in cleavage site of the mutant substrate whereby a value of 1 represents a complete shift in cleavage site while values <1 signify the proportion of cleavage occurring at the new site(s). Positive values denote a C-terminal shift (deeper into the TM) while negative values indicate an N-terminal shift (outward shift).

## EPR spectroscopy

Cysteines were introduced at positions 236 (TM5) or 247 (L5 loop) of GlpG in which the endogenous C104 was mutated to alanine. Proteins were expressed and purified as described previously (*Wu et al., 2006*), and labeled with 250 µM MTSL (Toronto Research Chemicals, Canada). Free probe was removed by gel filtration chromatography, followed by NiNTA affinity chromatography and washing for 2–3 days at room temperature. X-band EPR spectra of samples in 0.9-mm quartz capillaries were recorded at 37°C (310 K) on a Bruker EMXmicro spectrometer equipped with a PremiumX ultra low noise microwave bridge, a high-sensitivity ER4119HS resonator, and an ER4141VT temperature control unit (Bruker Biospin, Billerica, MA). Spectra were background corrected and subtracted, and normalized by the absolute number of spins in each sample as quantified by double integration using Xenon software (*Eaton et al., 2010*).

## CD spectroscopy

Thirty-two residue long peptides containing the entire TM sequence (with flanking lysines to increase solubility) were dissolved at 10–105 µM in 95% trifluoroethanol (Sigma, St Louis, MO) 1 mM DTT, or 10 mM Hepes pH 7 10 mM NaCl 1 mM DTT containing either 0.25% DDM or 1% SDS. Peptides were electrophoresed on 16% tricine gels to verify concentration and lack of aggregation, while peptides reconstituted into proteoliposomes were further examined by ultracentrifugation. Ellipticity and UV absorbance at 205 nm (to quantify peptide concentration accurately during scanning) were measured simultaneously at 25°C through a 0.2 mm path length cuvette in a Jasco J-810 spectropolarimeter (Jasco Inc., Easton, MD). Analyses were conducted by averaging 10 scans at 50 nm/min, background was determined and subtracted, and mean residue ellipticity was calculated.

## Rhomboid activity analysis in animal cells

*Drosophila* S2R+ and human HEK293 cells were transiently transfected with plasmids for the expression of GFP-tagged and/or Flag-tagged substrates and 3× HA-tagged rhomboid proteases (*Baker et al., 2006*). For non-substrates, we used GFP fused to the C-terminal most 99 residues from APP, GFP fused to full-length TGFα, and full-length *Drosophila* Delta (all constructs had a single Flag tag at their C-terminal ends). 24 hr after transfection, serum-free media was conditioned for an addition 18–24 hr. Media and cell samples were analyzed by quantitative westerns.

## Rhomboid activity analysis in bacterial cells

*E. coli* cells were transformed with two plasmids for the inducible expression of bacterial rhomboid proteases and Flag-tagged substrates, grown under double antibiotic selection, and induced with IPTG as described (*Urban and Baker, 2008*).

## Acknowledgements

We are grateful to all members of the Urban lab for scientific discussions and especially to Rosanna Baker for help with protein purification for EPR analysis, the Johns Hopkins Malaria Research Institute for use of their CD spectropolarimeter, and the *Drosophila* Genomics Resource Center for *Drosophila* clones and cell lines.

# Additional information

## Funding

| Funder | Grant reference number | Author |
| --- | --- | --- |
| Howard Hughes Medical Institute | 002137 | Syed M Moin, Sinisa Urban |
| National Institutes of Health | AI066025 | Syed M Moin, Sinisa Urban |
| Packard Foundation | 2007-31766 | Syed M Moin, Sinisa Urban |

The funders had no role in study design, data collection and interpretation, or the decision to submit the work for publication.

## Author contributions

SM, conducted in vitro proteolysis, bacterial culture and mass spectrometry experiments; SU, conducted in vitro proteolysis experiments, CD and EPR spectroscopy, and animal cell culture experiments, designed research, and wrote the manuscript; SM and SU contributed equally to the experimental work

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
