## [Author Response]

As requested, we have expanded our discussion of the predictive nature of the turn propensity scale (4th paragraph of the revised Discussion and a dedicated section in the Materials & Methods). This is an exciting topic for further study, and we have been careful to present a balanced description. On the one hand, we describe how we calculated the values (in the Materials & Methods) and point out, as noted by the Reviewers, that with such a high correlation it is likely to have predictive value as a tool for finding new rhomboid protease substrates. On the other hand, we caution that our correlation is based on a difference calculated between two substrates that are identical except at only 1 or 2 positions. As such, the relative *changes* in ‘turn propensity’ are highly controlled. A challenge described by the ‘turn propensity’ authors is that the propensity of the same residue differs when it is placed in transmembrane segments of different length, of differing sequence context, and depending on precisely where the residue is placed. As such, using the ‘turn propensity’ scale as an *absolute* value to predict rhomboid substrates from natural transmembrane sequences could prove to be challenging. Nevertheless, experimental refinement of the scale with rhomboid proteases directly is likely to improve its predictive accuracy even further.